# An Ornithologist's Guide for Including Machine Learning in a Workflow to Identify a Secretive Focal Species from Recorded Audio

**Ming Liu [1,†], Qiyu Sun [1,†], Dustin E. Brewer [2,*] , Thomas M. Gehring [2] and Jesse Eickholt [1]**

1   Department of Computer Science, Central Michigan University, Mt. Pleasant, MI 48859, USA
2   Department of Biology, Institute for Great Lakes Research, Central Michigan University,
    Mt. Pleasant, MI 48859, USA
*   Correspondence: dustinbrewer92@yahoo.com
†   These authors contributed equally to this work.

**Abstract:** Reliable and efficient avian monitoring tools are required to identify population change and then guide conservation initiatives. Autonomous recording units (ARUs) could increase both the amount and quality of monitoring data, though manual analysis of recordings is time consuming. Machine learning could help to analyze these audio data and identify focal species, though few ornithologists know how to cater this tool for their own projects. We present a workflow that exemplifies how machine learning can reduce the amount of expert review time required for analyzing audio recordings to detect a secretive focal species (Sora; *Porzana carolina*). The deep convolutional neural network that we trained achieved a precision of 97% and reduced the amount of audio for expert review by ~66% while still retaining 60% of Sora calls. Our study could be particularly useful, as an example, for those who wish to utilize machine learning to analyze audio recordings of a focal species that has not often been recorded. Such applications could help to facilitate the effective conservation of avian populations.

**Keywords:** ARUs; secretive marsh bird; Sora; deep learning; avian monitoring; passive acoustic monitoring

## 1. Introduction

Biodiversity is crucial for maintaining ecosystem function [1]. Therefore, wildlife population monitoring efforts are important for identifying population changes that could have broad ecological consequences. Monitoring efforts have revealed drastic population declines (e.g., in birds) [2] that appear to be caused by, and so could be at least partially reversed by, human activity. Some ecosystems, like wetlands, are particularly dynamic by nature and require efficient monitoring to quickly and effectively detect changes [3] that could endanger biodiversity and associated benefits, as well the system's intrinsic value.

Tracking wildlife population change, and responding with conservation efforts, has traditionally relied upon human observers. However, observer bias [4] and non-detection of cryptic species [5], among other sampling issues [6,7], present challenges that could reduce our ability to identify and respond to population declines. Thus, novel approaches are required to effectively monitor and so maximize our ability to conserve biodiversity. Especially for species that are acoustically conspicuous, like birds, the use of autonomous recording units (ARUs) for biological monitoring is becoming a viable alternative to using human observers [8]. ARUs can allow investigators to increase survey duration, decrease the number of site visits, acquire a permanent record of a survey period, and minimize observer bias. However, challenges exist for optimizing the use of ARUs, especially regarding efficient analysis of what is usually a large amount of audio data [9]. Analyses of audio data have often depended upon sub-sampling approaches, if data collection itself

was not curtailed to increase feasibility, to complete manual analysis of long recordings (e.g., [10]). Such approaches, however, are labor-intensive and could fail to detect species that only rarely produce an audio signal, such as secretive marsh birds [11].

Advances in automated approaches to identify birds from audio recordings have been chronicled in part by the BirdCLEF [12] series, which has organized annual bird identification competitions since 2014. The initial BirdCLEF competition focused on classifying more than 500 bird species from their vocalizations, recorded under varying conditions (e.g., use of different microphones and capture devices, recordings taken at different locations and different times of year, simultaneous vocalization and background noise) [12]. More recent BirdCLEF competitions have made use of audio recordings of over 900 bird species and evaluated methods on 10-min soundscapes from four distinct geographic regions [13]. The test soundscapes utilized in BirdCLEF competitions have contained dense acoustic scenes and cross-domain background noise (i.e., background noise that varies from collection point) and are characteristic of the challenges that a robust passive ARU pipeline must address in practice.

Early approaches by BirdCLEF participants to identify avian vocalizations from audio recordings utilized techniques such as template matching, decision trees, and support vector machines (overviewed in [14]). Template-detecting approaches, such as those facilitated by the R package 'monitoR' [15], allow users to create templates of the audio signatures of a target species and then search an audio survey for signatures similar to the templates. Templates have also been used as input to machine learning techniques (e.g., support vector machine, decision tree). This is done by employing the correlation between species-specific regions of interest in a spectrogram and a query spectrogram as input. Other low-level characterizations of audio that have been used as features include mel-frequency cepstral coefficients, energy in frequency bands, and pitch [16]. Initial BirdCLEF competition approaches often selected and engineered features to reduce the dimensionality of the input to classifiers (overviewed in [14]).

More recent BirdCLEF efforts to identify species from bird vocalizations have utilized deep learning [13,17]. Deep learning is typified by a large, layered composition of functions. Collectively, the layered composition of functions can be viewed as a network that transforms a numerical representation of the input into an output. Deep neural networks (DNNs) may be parametrized by tens of millions of parameters and the nested, layered composition of the functions which allow DNNs to capture complex patterns of patterns [18]. Between 2007 and 2022, DNNs have been applied to many domains and often outperform the prior state-of-the-art techniques without the need for extensive feature engineering or domain-specific preprocessing [18]. A specific type of DNN which makes use of convolutions, termed a deep convolutional neural network (DCNN), is well-suited to learn patterns of localized, semantic correlations of pixels present in images. The performance and utility of DCNNs also benefit from techniques such as transfer learning and data augmentation which enable the construction of large, deep networks with a small amount of data [19,20]. Due to these advantages, many audio classification tasks are reframed as image classification tasks. BirdNET, a tool for automated bird sound identification that has advanced due to BirdCLEF competition, exemplifies this approach as it is a DCNN operating on monochrome spectrograms [14]. To train the more than 27 million parameters in the BirdNET model, more than 1.5 million spectrograms were used in combination with extensive data pre-processing and data augmentation, and the resulting model was able classify 984 bird species by audio. Participants in recent iterations of BirdCLEF have relied heavily on DCNNs, making use of transfer learning and data augmentation [17]. To improve results in the BirdCLEF competition, many participants employ post-prediction filtering to adjust predictions based on the likelihood of a given species being present in a location [17].

Despite the promising results of DCNNs and approaches such as BirdNET, constructing a general, robust pipeline to identify bird species from surveillance data remains difficult. Effectively utilizing recordings with low signal to noise ratios, analyzing data

generated by lower quality recording equipment (such as those often deployed for monitoring purposes), and successfully identifying species which mimic others [14] are some challenges that still must be overcome. Additionally, recent studies on CNN-based models suggest that false negative rates are relatively high and caused by both faint bird sound and noise-masking by wind, insects, and weather [21,22]. These challenges make it such that investigators might be reluctant to use a general classifier such as BirdNET that has not been validated for their particular context.

We provide a workflow for investigators that wish to develop their own machine learning pipeline in order to analyze acoustic data. This workflow is especially useful when the target species has not been frequently recorded or when a more general classifier has not been trained or validated on the target species. By utilizing a combination of expert review and community-licensed machine learning software, we demonstrate how customized machine learning pipelines can be constructed to aid with the review of ARU recordings containing vocalizations of a rarely recorded bird species. Our approach strikes a balance between the time and domain expertise needed to develop an accurate classifier and the time needed for expert review. The overview presented herein, and the accompanying software, will be particularly useful for ornithologists with little or no computer science training who wish to recruit help from those with such training.

## 2. Materials & Methods

### 2.1. Study Species

We chose the Sora (*Porzana carolina*) as our focal species for several reasons. Soras are secretive marsh birds which, like other species in Rallidae, are rarely seen or heard. Thus, they are often difficult to detect by human observers when conducting surveys in the field and could potentially be better monitored by utilizing a workflow which involves ARUs for data collection and machine learning for analysis. Relatively few recordings of Soras exist and so existing machine learning pipelines may not accurately identify this species in ARU recordings. Consequently, the workflow that we present exemplifies steps that investigators could take to better monitor species, like Soras, which have been infrequently recorded and could be better monitored by leveraging the potential of ARUs and machine learning.

### 2.2. Data Collection and Curation

The data used in this work were gathered from both field recordings and xeno-canto (https://www.xeno-canto.org, accessed during March 2021; raw data for this study is at https://www.doi.org/10.17605/OSF.IO/RDJHY). Xeno-canto is an online repository of bird vocalizations that contains over 400,000 recordings that represent more than 10,000 bird species worldwide. Recordings from this dataset have been used in a variety of studies, including machine learning for automatic bird identification based on their vocalizations [21,23]. From xeno-canto, recordings containing sounds for the following species were collected: Pied-billed Grebe (*Podilymbus podiceps*), Least Bittern (*Ixobrychus exilis*), King Rail (*Rallus elegans*), Virginia Rail (*Rallus limicola*), Sora, Common Gallinule (*Gallinula galeata*), Marsh Wren (*Cistothorus palustris*), and Swamp Sparrow (*Melospiza georgiana*). These are the primary wetland bird species that occurred at our study sites and the expected bird vocalizations from our field recordings. The ID (i.e., xc-number), recordist, and URL for each recording from xeno-canto that was used for model training and evaluation are available at: https://www.doi.org/10.17605/OSF.IO/RDJHY. Table 1 lists the number of recordings from xeno-canto selected for each species.

**Table 1.** Number of recordings selected from xeno-canto by species and number of species-specific clips extracted.

| Species | Number of Recordings Used from Xeno-Canto | Number of Clips Extracted from Recordings |
|---|---|---|
| Common Gallinule | 10 | 310 |
| King Rail | 7 | 374 |
| Least Bittern | 12 | 1368 |
| Marsh Wren | 9 | 2500 |
| Pied-billed Grebe | 11 | 394 |
| Sora | 11 | 410 |
| Swamp Sparrow | 9 | 1048 |
| Virginia Rail | 11 | 588 |

In the field, seven ARUs (SM4, Wildlife Acoustics; Maynard, MA, USA) were deployed at Winous Point Marsh Conservancy (WPMC) in northwest Ohio and at Willow Slough Fish and Wildlife Area (WSFWA) in northwest Indiana to collect breeding bird vocalizations (calls and songs). The ARUs were placed in the field, at least 250 m apart, and attached to a post 1.2 m off the ground. The ARUs were set to record from 2 h before sunrise until 3 h after sunrise, as well as from 3 h before sunset until 2 h after sunset every day. In total, 10 h of recordings were made every day during our survey period, from mid-April to mid-June. All our recordings used in this analysis were made from 27 April 2020 to 12 June 2020. At WSFWA, sunrise ranged from 5:17 a.m. to 5:53 a.m. local time (central). At WPMC, sunrise ranged from 5:55 a.m. to 6:49 a.m. local time (eastern).

The ARU recordings were saved on secure digital (SD) cards as wav files at a sample rate of 24,000 hertz. Using version 2.3.3 of Audacity® recording and editing software (https://www.audacityteam.org/, accessed during March 2021), we changed the 'project rate' of all recordings to 22,050 and resampled at 22,050 hertz so that xeno-canto and ARU recordings ultimately had the same sample rate. From the ARU recordings, additional sounds were selected for Pied-billed Grebe, Sora, and Swamp Sparrow. Additionally, several non-bird sounds from field recordings were also selected and included in the dataset. We provide the number and type of clips extracted from instances of a target sound recorded by ARUs (Table 2).

**Table 2.** Number of recordings used from ARUs by species/type and number of clips extracted.

| Species/Type | Number of Instances Used from ARUs | Number of Clips Extracted from Instances |
|---|---|---|
| Variable Noise | 44 | 98 |
| Pied-billed Grebe | 3 | 34 |
| Sora | 103 | 286 |
| Swamp Sparrow | 8 | 10 |

Prior to model training, the acoustic signals in the wav files were transformed to spectrograms. The use of spectrograms has proven to be a successful approach for automatic bird species identification based on their vocalizations [21,22,24]. Spectrograms maintain abundant information and can be used for bird species identification based on the patterns of frequency and intensity throughout a vocalization's duration. The x-axis, y-axis, and the color of the spectrograms represent duration, frequency, and amplitude of vocalization components, respectively [21,22,24]. Bird sounds are different among species and from ambient noise, and these differences can be detected as patterns in acoustic duration, intensity, and frequency on spectrograms (e.g., Figure 1). To convert the acoustic signals to

spectrograms, the audio files were first split into overlapping clips 1.5 s in length with an overlap of 1 s. This was accomplished with Sox version 14.4.2. Sox was then used to create a raw spectrogram measuring 256 by 256 pixels for each clip.

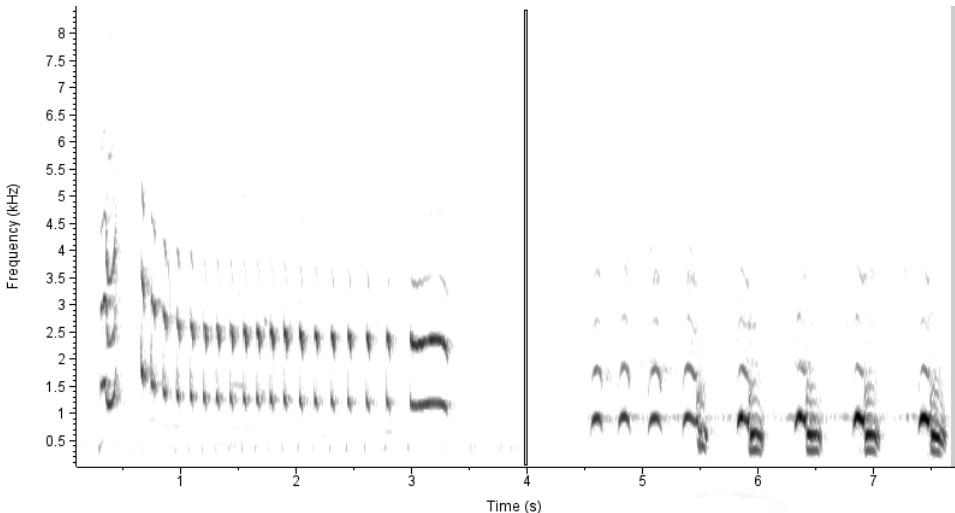

**Figure 1.** Spectrograms representing the most common Sora call encountered during our study ('whinny', on **left**) and a call of a non-focal marsh bird species, the Pied-billed Grebe (on **right**). Note that each of the calls span approximately 3 sec and so would have been split into multiple 1.5 s clips during analysis.

As the aim of this work was to identify Sora calls in audio recordings, the data were labeled as Sora and non-Sora. The 1.5 s clips were assigned the same label as their corresponding source file. Training and validation datasets were created by randomly splitting lists of Sora and non-Sora audio files into two bins, with roughly 80% of the files being used for model training. When dividing the data, care was taken to ensure that all clips stemming from the same source file were assigned to the same dataset. Finally, the spectrograms for the training data were visually inspected; spectrograms labeled as Sora, but which did not contain a visible signal, were discarded.

To characterize a classifier's effectiveness at identifying Sora in situ, two additional unbalanced datasets were constructed. The first stemmed from a 195-min recording collected from an ARU deployed on 28 April 2020 in known Sora habitat. The second stemmed from a 240-min recording deployed on 27 April 2020 at the same site. Given that these recordings occurred during a migratory period, high densities of Soras (e.g., >10 within range of the ARU) could have been recorded, though the exact number of individuals present was unknown. Both recordings were sliced into overlapping clips 1.5 s in length. Clips from the 28 April 2020 recording were used as an additional, 'extended' validation dataset that served to fine-tune our algorithm before the final dataset was tested. Clips from the 27 April 2020 recording were used as the final evaluation dataset. Given that Sora vocalizations were infrequent, most of the audio captured from the ARU did not contain Sora vocalizations (i.e., the distribution of the labels was unbalanced and highly skewed towards non-Sora). These imbalanced datasets better represent the conditions in which the classifier would be used. Ranges of the longer clips containing Sora vocalizations were identified by expert analysis. Any 1.5 s clip that overlapped an identified range by 0.5 s was labeled as a Sora vocalization. In the additional validation dataset, 732 of 23,404 clips were labeled as Sora (3.1%) and in the final evaluation dataset, 312 of 28,796 clips were labeled as Sora (1%).

### 2.3. Model Creation

The Keras library [25] was used to train and evaluate a DCNN. The model architecture employed was based on the Inception ResNetV2 [26], trained on imagenet and provided

through the Keras API. The output of the base model was passed through a global average 2D pooling layer and two additional fully connected layers with 2048 and 256 relu nodes, respectively. The dense layers made use of l2 regularization and were interspersed with two layers of dropout. The final output of the model was one node, making use of a sigmoid activation energy.

The model was trained for 34 epochs with the Adam optimizer at a learning rate of $2 \times 10^{-5}$ with binary cross entropy as the loss function. The spectrograms were augmented with random horizontal shifts of up to 5% and a balancing factor was applied to both the training and validation data during training to ensure that roughly an equal number of Sora and non-Sora clips were selected in each epoch of training. On a balanced presentation of the training and validation data, the model achieved an accuracy of 96.3% and 91.2%, respectively.

Several model architectures and training procedures were evaluated but an extensive model tuning and selection process was not performed.

*2.4. Model Evaluation Procedures and Metrics*

The output of our model was a real value in the range 0 to 1, with values closer to 0 indicating a greater probability that the input clip did not contain a Sora vocalization and values closer to 1 indicating a greater probability that the input clip did contain a Sora vocalization. To map the output of a model onto the two discrete labels, a decision threshold was used. Model outputs greater than the decision threshold were mapped to Sora and those less than the decision threshold were mapped to non-Sora. The default decision threshold for evaluation during training was 0.5.

The principal metrics used to evaluate binary classifiers are accuracy, precision and recall. Each prediction made by the classifier can be categorized as a true positive (TP), true negative (TN), false positive (FP), or false negative (FN). For this work, the Sora label was considered the positive label, so a true positive was a Sora clip correctly predicted to be a Sora vocalization. A false positive was a non-Sora clip that was incorrectly predicted to be a Sora vocalization. True negative and false negative are similarly defined as correctly and incorrectly predicting non-Sora calls, respectively. Accuracy is defined as TP + TN/(TP + FP + TN + FN). Precision is defined as TP/(TP + FP) and recall is defined as TP/(TP + FN). Precision and recall were calculated for the extended, unbalanced datasets making use of different decision thresholds.

Before evaluating the model's performance on the longer recordings an additional post-processing step was performed. The post-processing step recalculated the model's output for a clip using the weighted average of the clip along with the model's output, m(i), for the preceding and proceeding clips. More specifically, the model's output for the i-th clip, extracted from a longer recording, was reweighted as 0.175 × m(i − 1) + 0.175 × m(i + 1) + 0.65 × m(i). Since the extracted clips overlap by 1 s, this reweighting scheme leverages evidence of a Sora call contained in the neighboring clips and also provides a modest ensembling effect. The python script for this study is at https://www.doi.org/10.17605/OSF.IO/RDJHY.

Given that an aim of this work was to reduce the amount of surveillance data needing manual inspection while maximizing identification of Sora calls, an additional performance metric—'Sora calls recovered'—was employed. When culling surveillance data, it is important that portions of a recording that contain a Sora clip are retained to allow for later expert identification. For the unbalanced validation dataset and unbalanced final evaluation dataset, there were 67 and 43 Sora calls, respectively. Some calls were longer than others, and all calls were split over several 1.5 s clips. 'Sora calls recovered' differs from 'recall' in that recall is a performance metric applied to clips, whereas Sora calls recovered is a performance metric applied to Sora calls. Recall measures the fraction of clips that contain a Sora vocalization which were predicted to contain a Sora vocalization. Similarly, Sora calls recovered measures the fraction of Sora vocalizations in a longer recording which was

retained for manual inspection by an expert reviewer. A Sora call was considered recovered if at least one of its 1.5 s constituent clips was correctly predicted to be a Sora vocalization.

## 3. Results

Of the clips extracted from the longer ARU recordings, the DCNN model achieved accuracies in the range of 0.53 to 0.94. Tables 3 and 4 list the precision and recall of Sora vocalizations, along with the overall accuracy of the predicted class for clips (i.e., Sora or non-Sora) across a range of decision thresholds. As the distribution of classes is highly skewed towards clips without Sora vocalizations, precision is a better indicator of the classifier's performance (i.e., with ~97% of all clips being non-Sora, a naive approach which always predicted non-Sora would achieve an accuracy of 97% but would not identify clips containing Sora vocalizations).

**Table 3.** Accuracy, precision, and recall of DCNN predictions for the extended validation dataset.

| Decision Threshold | Accuracy | Precision | Recall |
|---|---|---|---|
| 0.5 | 0.62 | 0.04 | 0.49 |
| 0.6 | 0.68 | 0.04 | 0.42 |
| 0.7 | 0.74 | 0.04 | 0.35 |
| 0.8 | 0.81 | 0.05 | 0.27 |
| 0.9 | 0.89 | 0.05 | 0.16 |
| 0.95 | 0.93 | 0.06 | 0.09 |

**Table 4.** Accuracy, precision, and recall of DCNN predictions for the final evaluation dataset.

| Decision Threshold | Accuracy | Precision | Recall |
|---|---|---|---|
| 0.5 | 0.53 | 0.01 | 0.64 |
| 0.6 | 0.61 | 0.01 | 0.57 |
| 0.7 | 0.69 | 0.02 | 0.47 |
| 0.8 | 0.78 | 0.02 | 0.39 |
| 0.9 | 0.89 | 0.02 | 0.23 |
| 0.95 | 0.94 | 0.03 | 0.13 |

Also calculated were the number of Sora calls recovered and the reduction in data for the extended validation and final evaluation datasets (Tables 5 and 6). These tables represent the extent to which the surveillance data could be culled and coverage of Sora calls across different decision thresholds. At a decision threshold of 0.8, a human expert would only need to listen to, and/or view spectrographically, about 67.5 min of culled clips to recover 60% of Sora compared to 195 min to examine the entire longer clip (See Table 5). The recall of Sora vocalizations can be increased by reducing the decision threshold. However, the increase in recall comes at the cost of decreased precision and more effort by a human expert to filter out false positives.

**Table 5.** Number of culled clips (of variable length), total length of clips, and percentage of Sora vocalizations recovered by culled clips on the extended validation dataset.

| Decision Threshold | Number of Culled Clips Extracted for Inspection | Total Length of Culled Clips for Inspection (in Minutes) | Percentage of Sora Calls Recovered |
|---|---|---|---|
| 0.6 | 627 | 161.92 | 91 (61/67) |
| 0.7 | 659 | 121.42 | 79 (53/67) |
| 0.8 | 480 | 67.5 | 60 (40/67) |
| 0.85 | 353 | 43.67 | 43 (29/67) |
| 0.9 | 195 | 22.08 | 22 (15/67) |
| 0.95 | 59 | 6.08 | 6 (4/67) |

**Table 6.** Number of culled clips (of variable length), total length of clips, and percentage of Sora vocalizations recovered by culled clips on the final evaluation dataset.

| Decision Threshold | Number of Culled Clips Extracted for Inspection | Total Length of Culled Clips for Inspection (in Minutes) | Percentage of Sora Calls Recovered |
|---|---|---|---|
| 0.6 | 652 | 234.4 | 86 (37/43) |
| 0.7 | 804 | 186.3 | 81 (35/43) |
| 0.8 | 713 | 111.5 | 63 (27/43) |
| 0.85 | 528 | 69.8 | 49 (21/43) |
| 0.9 | 323 | 35.5 | 33 (14/43) |
| 0.95 | 81 | 8.4 | 14 (6/43) |

## 4. Discussion

The value and appeal of a robust, end-to-end solution to efficiently process passive ARU data for avian surveillance are well-documented and are the impetus behind community efforts, such as the BirdCLEF competition series, to document and advance the discipline. While much progress has been made in the past 10 years, including in regard to monitoring secretive marsh birds [27], challenges remain. Notable challenges include domain-specific background noise, unbalanced target distribution, varying audibility of vocalizations, and weakly labeled training data.

The aforementioned challenges, coupled with limited amounts of labeled data for some species, suggest that fully automated pipelines to process ARU data will not be viable in the near term. In our study, this reality is exemplified by the poor performance of the DCNN in identifying Sora calls on the longer, unbalanced ARU recordings. The precision and recall on the longer ARU recordings were low and while extensive model selection and refinement were not attempted, it is unlikely that those efforts would have had a significant effect on the model's performance. The high accuracy of the model on the limited, balanced data used for initial training and initial validation indicates that the model is not generalizing very well. This is likely due to the limited amount of labeled training data. The paucity of Sora vocalizations in longer ARU recordings increases the impact of a model's false positive rate and this further complicates attempts to fully automate ARU data processing in this setting.

### 4.1. Culling Surveillance Data to Identify Vocalizations of Rarely-Recorded Bird Species

Despite the DCNN's poor performance at identifying Sora vocalizations on longer, unbalanced recordings (i.e., the precision of the model's predictions is very low, with a large number of false positives), the results do support a semi-automated workflow (Figure 2) that could significantly reduce the amount of data that would need expert review. As

shown in Tables 5 and 6, the DCNN is capable of recognizing parts of Sora vocalizations and while it cannot correctly identify an entire vocalization, areas that are predicted to contain Sora vocalizations can be marked and extracted for manual review. Expert review of the culled surveillance data would remove the false positives. Additionally, if large amounts of surveillance data are available, a bootstrapping procedure in which a DCNN is trained using the labeled data on hand and then additional vocalizations are identified through manual inspection of the culled data could be implemented. As an example, on the final evaluation dataset roughly 15% of the Sora vocalizations would have been recovered by inspecting only 3.5% of a 4-h recording (8.4/240 min; See Table 6). An initial model applied to several days of surveillance data could yield many additional vocalizations that could augment the training data and lead to better models. These better models could again be applied to surveillance data to identify more Sora vocalizations.

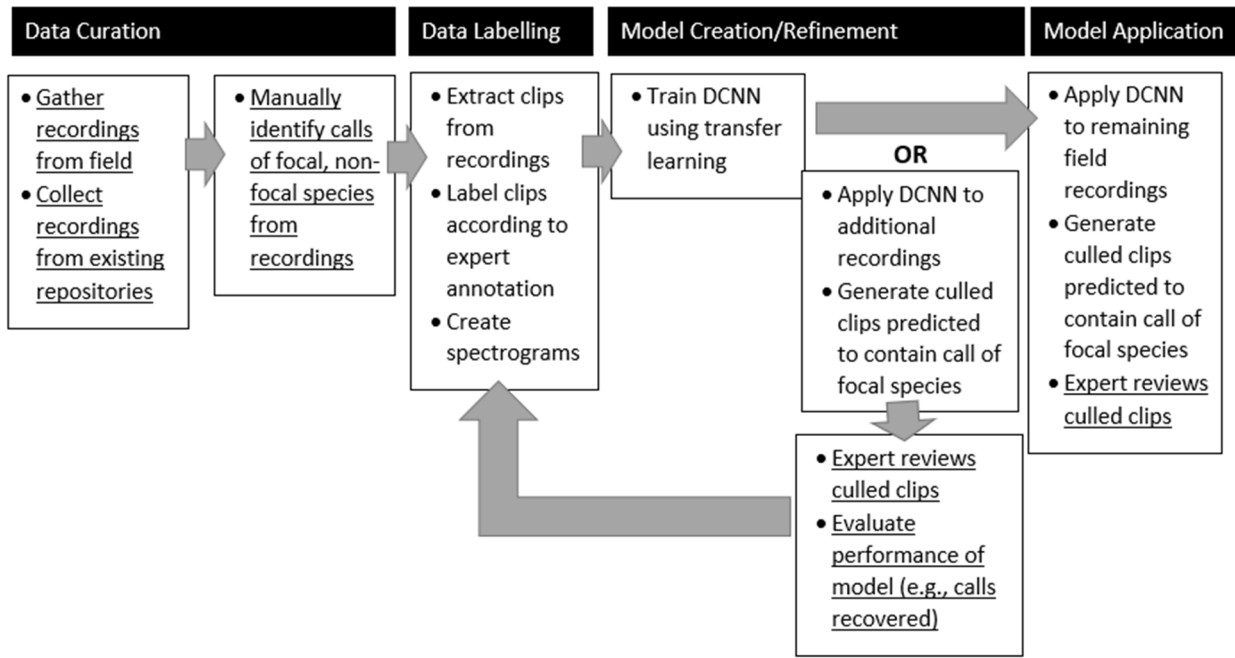

**Figure 2.** An overview of the workflow that we used to identify Sora calls from audio recordings. The steps that are underlined were completed by an ornithologist and the other steps by computer scientists.

In general, the construction of a semi-supervised pipeline to process ARU recordings of vocalizations of a rarely recorded bird species can be constructed quickly and with community-licensed tools and data. Ornithologists can collect and label audio clips containing vocalizations of bird species of interest. Audio clips can also be obtained from xeno-canto or recordings collected in the field. To limit the effect of cross-domain background noise, it is preferable to include some data collected at the surveillance site. A machine learning library such as Keras can be used to train a DCNN with the limited labeled data and transfer learning. Extensive model selection and evaluation is not required. The DCNN can then be applied to the surveillance data (e.g., multiple weeks of ARU data) to cull the data for manual inspection. If desired, additional training data can be garnered through manual inspection and another model can then be trained and applied to the surveillance data. The number of iterations needed would depend on the needs of the ornithologist and the study. With this workflow, less effort could be spent on building robust models that generalize too many settings; more effort therefore would be spent on inspecting the results of the models.

### 4.2. Techniques to Build More Generalizable Data Processing Pipelines

One means to combat the challenges in developing a general, robust avian classifier (e.g., domain-specific background noise, unbalanced target distribution, varying audibility

of vocalizations and weakly labeled training data) is to build task-specific classifiers, as described in the previous subsection. Other means include pre-processing of the training data and post-processing of a model's output, and in recent years a number of clever schemes have been developed. A no-call binary classifier that identifies portions of audio that do not contain bird vocalization can serve to weight weakly labeled data for training or to filter predictions. Common to pipelines evaluated in BirdCLEF competition are specific post-processing techniques such as filtering by time and location of recording, boosting, and ensembling [17]. Increasing the confidence of predictions for the most common bird species or removing species with which a model is known to overpredict will decrease false positives [28]. While these approaches provide a boost in performance, some of them work contrary to the aims of studies such as ours due to reduced likelihood of detecting rarely occurring species. Additionally, such approaches might not be practical for automated avian surveillance due to their computational expense.

To combat differences in signal strength between the more-focused recordings used for training and longer recordings typical of surveillance collection (e.g., surveillance data may contain many vocalizations at varying distances, long periods of no-calls, different background sounds), data augmentation can be utilized to inject noise into the training clips. The injected noise stems from annotated soundscapes known to be devoid of bird vocalizations [29] or white or pink noise [28]. This process results in models that are more effective in low signal-to-noise settings [29]. Randomly introducing noise into the training data allows the model to learn to cope with the noise. This approach is typical of deep learning in which domain-specific pre-processing (e.g., denoising audio clips to increase vocalization-to-noise ratio) is not needed and the model is able to learn effective transformations to characterize the raw input for the classification task at hand [18].

### 4.3. Applicability of More General Avian Classifiers

It is often unknown how larger, more general pipelines will perform for smaller, species-specific tasks. A trained deep neural network represents a mapping, from inputs (e.g., spectrograms) to labels (e.g., bird species), that is parameterized by the weighted composition of many functions. Neural networks are often criticized as "black boxes" since it is difficult to interpret the weights that parameterize a trained model [30]. The learned rules that a deep neural network embody can be evaluated by using known, novel data to estimate its performance. However, the rules cannot be meaningfully understood to gauge how they might apply to other classification tasks (e.g., general species classifier to a more-specific species classifier, one environment to another environment). Thus, ornithologists may be reluctant to use an existing general avian vocalization classifier if it is unclear if the classifier would perform similarly on a single-species identification task applied to data collected in a different environment, with different equipment. To assess a model's suitability for a specific monitoring program, an investigator would need to collect and manually label vocalizations and then assess the model using the new curated evaluation dataset.

Investigators might be better served by developing their own task- and species-specific classifier rather than using an existing avian classifier, especially if existing classifiers were found to be insufficient to accomplish their objectives. In BirdCLEF 2021 it was noted that many participants used standard, pre-trained DCNNs as the backbone to their methods [31]. By utilizing transfer learning and data augmentation, specialized task or species classifiers can be trained and evaluated with a limited amount of labeled data, as evidenced in this work. Both the evaluation of an existing general avian vocalization classifier and the creation and evaluation of a more-specialized classifier require the construction of a new labeled dataset. The specialized classifier can then be used to cull surveillance data for later expert review. For a given problem, the time, effort, and expertise needed to train and evaluate a highly accurate, task-specific classifier may be significantly more costly than constructing a simpler classifier and relying on expert review of the results. The latter approach has the benefit that the results are further validated through expert review.



## 5. Conclusions

Efficient and effective monitoring approaches are required to identify the conservation needs of organisms and then to implement actions to slow or reverse those declines. Machine learning could revolutionize the way that birds are monitored by reducing the amount of time required to review ARU recordings. Our method could, for example, improve the feasibility of incorporating automation when attempting to identify a very rare (or presumed extinct) bird species in audio recordings compared to previous automation applications [32]. There are situations when detecting even one focal species occurrence could be useful, regardless of the recall rate of the utilized approach. For example, our approach could be useful even with the low recall rate that we observed if it helped to establish the presence of a particularly rare species of conservation concern.

Despite the great potential for machine learning to improve monitoring efforts, many ornithologists lack the technical knowledge required to effectively use machine learning themselves to identify when focal species occur in audio recordings. Clearly, additional work is required to continue improving the effectiveness and accessibility of workflows that involve machine learning in the context of avian monitoring. Our study could help to provide a collaboration roadmap for ornithologists and computer scientists interested in achieving this goal to ultimately advance the efficiency and effectiveness of avian monitoring and conservation programs.

**Author Contributions:** Conceptualization: M.L., Q.S., J.E. and D.E.B.; methodology, M.L., Q.S., J.E. and D.E.B.; software, M.L., Q.S. and J.E.; validation, M.L., Q.S. and J.E.; formal analysis, M.L., Q.S. and J.E.; investigation, M.L., Q.S., J.E., D.E.B. and T.M.G.; resources, J.E.; data curation, J.E., writing–original draft preparation, J.E., D.E.B., M.L. and Q.S.; writing–review and editing, M.L., Q.S., J.E., D.E.B. and T.M.G.; visualization, M.L., Q.S., J.E. and D.E.B.; supervision, J.E. and T.M.G.; project administration, J.E.; funding acquisition, J.E. and T.M.G. All authors have read and agreed to the published version of the manuscript.

**Funding:** This research was funded in part through a grant from the U.S. Fish and Wildlife Service, Division of Bird Habitat Conservation (grant award: F19AP00330).

**Data Availability Statement:** All data and scripts are available at: https://www.doi.org/10.17605/OSF.IO/RDJHY.

**Acknowledgments:** We thank M. Schoof for providing equipment and field support for our study and Winous Point Marsh Conservancy for letting us conduct research on their property. This research was supported by Central Michigan University's Earth and Ecosystem Science PhD program, the Department of Biology, the Institute for Great Lakes Research, and the College of Science and Engineering. This is contribution number 170 of the Central Michigan University Institute for Great Lakes Research.

**Conflicts of Interest:** The authors declare no conflict of interest.

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
