# Peer review of "An Ornithologist’s Guide for Including Machine Learning in a Workflow to Identify a Secretive Focal Species from Recorded Audio"

_remotesensing, doi:10.3390/rs14153816_

Round 1

Reviewer 1 Report

The authors developed a machine learning tool to identify the vocalizations of a cryptic bird in passively recorded audio, and described their workflow. The authors present their study as a “roadmap” for ecologists (402) and I think that could be a reasonable statement once they address the issues outlined below. The first of these is that they don’t actually provide a literal roadmap! A conceptual figure that shows the workflow with boxes and arrows would be a valuable addition to the process, and color-coding the boxes to highlight which task an ecologist could reasonably complete and which would require someone with substantive computer science skills would be the most valuable part.

The content from 52-96 feels jumbled. It is generally an overview of the history and mechanics of audio event classification, but the order of the paragraphs does not feel coherent and the ideas do not seem to flow.

Why only train on two classes, Sora and non-Sora? Is there evidence to suggest that this could lead to better performance? To me it seems like a major missed opportunity. Generating population data about non-target species is a major potential benefit of passive acoustic surveys + ML tools – yet the authors did not take this opportunity. Why? I would strongly suggest to anyone considering creating a classifier that it encompass as complete a species list as reasonably possible. Users can simply filter their results for the species of interest.

The description of the validation datasets (195-208) is lacking. Why were these long recordings selected? The dates do not answer that question at all. Were these particular ARUs deployed at a location at which Sora were known to vocalize (or at which the researchers expected to record vocalizations)? Such deployments are – in my opinion – and extremely important detector testing best practice that should be highlighted, but it is not clear that this occurred here. Also, does this mean that detector performance was essentially measured against the vocalizations of just one individual or pair? That would seem to miss lots of relevant variation in target signals. First, many bird species can be differentiated at an individual level on the basis of consistent differences in highly stereotyped vocalizations. Second, unless the birds are highly mobile, their vocalizations during those periods were likely to come from roughly the same distance and feature roughly the same levels of background noise and interference. I’m sympathetic that the cryptic nature of this species makes it difficult to amass enough examples for both training and independent validation, but the approach that seems to have been used here would lead to a non-representative validation dataset.

Change Table 3, 4, 5, 6 into Figures. This will make performance much easier to quickly understand.

An important point that is missed entirely is the fact that not all vocalizations need to be recovered. From a conservation perspective, the critical question is whether a species is present at a given location. Simplistically, this merely requires a single true positive for a given recorder. For example, Wood et al. (2019 – Ecological Informatics) note this point by measuring detector recall at the level of the call and the level of the bout of calling. Overall, this means that low recall does not automatically mean that the detector has performed “poorly”. There is a lot of possible nuance to recall that is being overlooked. (A corollary of my point here is that in a semi-automated or fully manual workflow, a human analyst could stop validating additional results once the target species has been identified at a site – which could lead to a big time/effort savings.)

As an ecologist who’s familiar with ML techniques, collecting and labeling a dataset seems easier than developing my own ‘net and is not an unreasonable expectation when considering a tool in a new context (375-377). In fact, I would need to do collect and label vocalizations in order to make my own ‘net anyways. …and I see that this is noted in lines 383-5. Overall, I do not find this a compelling reason for someone to develop their own model rather than using an existing one. To me, the reason to make a new model is that testing of an existing one revealed insufficient performance for my objectives.

Minor Comments

30-31: It would be nice to include rationales for conservation that go beyond the purely utilitarian

45: machine learning tools are not without bias. However, that bias is much more readily measured and, by using the same observer” for all sites, the bias is consistent.

48: I have seen people subsample continuously recorded data, but, in my opinion, what’s worse is only recording tiny amounts of data periodically (eg, 5 minutes of every hour) because that’s all they can manually review. This prevents them from reanalzying a much larger dataset when better tools, like machine learning, to become available.

49: “as illustrated by one study” is an awkward way to make this point. Parenthetical mentions would be better – “(e.g., [10])”.

97: "BirdNET"

150: "apart from each other" is redundant

166-7; 180-1; 270-1; and probably elsewhere: it is usually best to reference tables (and figures) parenthetically

172: generalize the statement by replacing "bird species" with "acoustic event"

227-9: the output metric is a continuous variable that’s being used to describe the probability of a binary outcome. Therefore, closer to 1/1 should be described as “more/less likely to/not to contain a Sora vocalization”

248: “i-th” might be easier to understand than “ith”

252: including the word “stated” creates a strange impression that there were perhaps “unstated” aims of the project.

257: I would like to see a sentence explaining how this is different from recall

292: “This is exemplified in this work by…” is confusing. Too much “this” and not enough specificity.

357: delete “By”

363: I’m confused by “additional”. “Additional” to what? Introductory sentence of a new section needs to be able to stand alone. 

Reviewer 2 Report

ARU is abbreviated in the abstract section. Please use only ARU (introduction section on page 1) in the remaining section.

Reviewer 3 Report

Dear authors,

Congratulations on your research describing a workflow model demonstrating how machine learning can assist ornithologists/ecologists in analysis of audio recordings.

A few point for your consideration:

Title. Should the word Focal be Vocal? this applies to all uses of the word focal in the paper.

Methods: Study "organism" change to study "species". Also it needs to be stated that recordings were done during the breeding season. Breeding vocalisations? need to clarify for ecologists.

Methods: Include the ARU brand/model as this is critical to the quality of the recording. 

Figure 1. Map needs more information, state etc. or leave out all together as doesn't add to the paper.

Figure 2. X axis requires labelling with temporal scale and y axis frequency.

Table 3 and Table 4. Precision is very low in this work. This reduces the efficacy of the method in a real world situation. Never the less, the paper is interesting in its attempt to bring CNNs down to a scale useable by a reasonably competent ecologist.

Round 2
